# The associations between resilience, self-care, and burnout among medical students

**Keren Michael** [1]*, **Dana Schujovitzky**[2,3], **Orit Karnieli-Miller**[2]

**1** Department of Human Services, Max Stern Yezreel Valley College, Yezreel Valley, Israel, **2** Faculty of Medical and Health Sciences, Department of Medical Education, Tel Aviv University, Tel Aviv, Israel, **3** Department of Pediatrics, Meir Medical Center, Kfar Saba, Israel

* kerenmi@yvc.ac.il

**Data Availability Statement:** All relevant data are within the manuscript and its Supporting Information files I attached all the data from the participants in a pdf file

## Abstract

Burnout is a work-related stress syndrome with substantial consequences for patients, physicians, and medical students. Personal resilience, i.e., the ability to bounce back and thrive despite challenging circumstances, and certain practices, such as self-care, may protect individuals from burnout. However, limited information exists on the complex relationships between resilience, different self-care practices, and burnout. Understanding these associations is important for designing efficient interventions within medical schools. Therefore, the present study examined the direct and indirect associations through a cross-sectional study among 95 fourth-year medical students. Self-reported questionnaires measured resilience, self-care dimensions (stress management, spiritual growth, interpersonal relations, health responsibility), and burnout dimensions (emotional exhaustion, depersonalization, personal accomplishment). Data were analyzed via IBM-SPSS and PROCESS-macro. The main results demonstrated that self-care mediated the associations between resilience and burnout: stress management and interpersonal relations mediated the associations with emotional exhaustion, while spiritual growth mediated the association with personal accomplishment. These results highlight that medical students' resilience may encourage self-care behaviors, thus decreasing levels of the burnout dimensions of emotional exhaustion and personal accomplishment. Developing curricula that enhance students' resilience through applying self-care techniques in stressful situations may reduce the negative impact of burnout in healthcare.

## Introduction

Burnout is a dynamic, stress-related process caused by a combination of high workload and low coping resources. Dimensions of burnout include emotional exhaustion––a lack of energy and feeling worn out; depersonalization––a negative approach to others and treating them as objects, and decreased sense of personal accomplishment––feelings of incompetence and lack of achievement or productivity at work [1]. Burnout is common in members of the healthcare professions, including nurses, physicians, and medical students, who are involved in caregiving and intensive interaction with people, often with a tendency to put others in the center,

**Funding:** This study was financially supported by the Israel National Institute For Health Policy Research in the form of a grant (234-17) received by KM and OKM. This study was also financially supported by the Dr. Sol Amsterdam & Dr. David P. Schumann Chair in Medical Education in the form of a grant (2024) received by OKM.

**Competing interests:** The authors have declared that no competing interests exist.

sometimes compromising their own well-being [2]. Burnout has severe personal consequences. For example, medical students' burnout was associated with low sleep quality [3], alcohol abuse/dependence [4], depression [5], and suicidal ideation [6]. Burnout has negative professional implications for medical students, including academic problems [3], decreased altruistic professional values, suboptimal patient care, and medical errors [6].

A recent meta-analysis [7] involving 42 studies and 26,824 undergraduate (pre-intern) medical students reported an overall burnout prevalence rate of 37.23%. Prevalence of emotional exhaustion was 38.08%, depersonalization was 35.07%, and personal accomplishment was 36.85%. When comparing medical students to age- and education-matched population control samples, medical students were more likely to be burned out [6,8,9], with an increased burnout percentage during years of medical school [5,10].

Despite the high incidence of burnout among medical students and its problematic consequences [11], not all medical students experience burnout, and some manage to overcome it [12]. One personal ability that may protect individuals from burnout is resilience [13]. Resilience is the capacity to bounce back and adapt positively when faced with adverse, challenging, and threatening situations [14,15]. Resilient individuals are usually optimistic, with high coping competence and high tolerance, which are related to better physical and mental health [16]. Two substantial components are required for resilience to be expressed: adversity and positive adaptation in the face of that adversity. Adversity is usually a challenging event with a high likelihood of a problematic outcome; however, positive events, such as acceptance to medical school and work promotion, may also pose challenges and require resilience. Positive adaptation is a behavioral manifestation of social competence or of successfully meeting stage-salient developmental tasks [14].

Although the role of resilience in preventing burnout has been extensively studied, and the negative association between resilience and burnout is well-established [17–19], the intuitive connection between resilience and burnout has become an obstacle since a considerable number of researchers measured resilience as a "lack of burnout," notwithstanding the distinction between the two concepts [12,20]. According to Strümpfer [21], underpinning the concept of resilience are some psychological factors that enhance the resistance to burnout. These psychological factors include engagement, meaningfulness, subjective well-being, positive emotions, and proactive coping. *Engagement* is a fulfilling work-related cognitive-affective state that is characterized by vigor, dedication, and absorption. *Meaningfulness* describes individuals' ability to perceive themselves as necessary and valuable. *Subjective well-being* describes both the sense of value and satisfaction from the person's daily life and enjoyment of life. *Positive emotions* describe a mental state that, in the short term, is conducive to increasing the conscious broadening of desirable thoughts and, in the long term, to performing effective and valuable activities [22]. *Proactive coping* describes people's ability to identify situations that are likely to cause stress and to behave in a way that will prevent them from worsening and even lead to posttraumatic growth [23]. These psychological factors of resilience can be nurtured by developing various individual and environmental behavior components, such as a sense of control, a sense of meaning in the life routine, and the presence of a mentor attuned to a worldview of competence and confidence [24].

During their taxing years in medical school, and especially during their clinical years, medical students are exposed to high prevalence of stress, distress, and adverse life events, such as high academic demands, economic difficulties, situations of helplessness, and moral and ethical dilemmas [25,26]. Additional distress relates to the tendency among healthcare professionals, including medical students, to focus on the care of others, sometimes at the cost of self-neglect [2] and lack of self-care. In view of this tendency, the present study focused on

examining how medical students' self-care practices during their stressful clinical years of training were related to burnout.

## Self-care practices and their associations with resilience and burnout

Self-care refers to the "ability to refill and refuel oneself in healthy ways" [27] or to having a caring attitude toward the self [28]. Self-care is a pattern of activities initiated and performed by individuals to either preserve or enhance their well-being while promoting their health potential [29]. Researchers have proposed various ways of promoting self-care: striving for balance between work and personal life, seeking sources of support, exercising, and getting adequate sleep [30,31]. Integrating such activities into a daily lifestyle improves individuals' health and quality of life [32]. A recent literature review on self-care [33] suggested that fostering self-care in cognitional, spiritual, emotional, social, and physical areas can promote professionals' upward spiral of well-being and help to avert the downward spiral of stress and burnout.

Indeed, studies indicate a negative association between self-care and burnout among healthcare professionals [34,35]. Moreover, studies used self-care interventions to deal with burnout. For example, in a study with nurses whose burnout was measured before and after a self-care intervention, a significant post-intervention decrease was found in emotional exhaustion and depersonalization [20]. However, to the best of our knowledge, limited empirical studies have examined the associations between the different self-care dimensions (stress management, spiritual growth, interpersonal relations, and health responsibility) and burnout among medical students. To design educational interventions, it is necessary to assess and identify which self-care dimensions are particularly helpful for preventing burnout. As these self-care dimensions have the potential to prevent burnout among medical students, we will explore them and elaborate on them in this manuscript.

*Stress management* involves allocating psychological and material resources to reduce or control stress [36]. Stress management is expressed in leisure activities, physical exercise, or adequate rest, as well as in individuals' ability to balance work and leisure, to use specific stress-controlling techniques, and to adjust the pace of life to their physical and psychological needs [37]. Stress management techniques such as mindfulness meditation practices, relaxation, and mind–body interventions may help students deal with pressure, reorganize their lives, and control their time and energy resources [25]. In a study among nurses in Hong Kong, stress management was negatively correlated with depersonalization and positively correlated with personal accomplishment [37]. Although the literature demonstrates implementation of various stress management techniques and study of their influence either on resilience (e.g., [12]) or on burnout (e.g., [38]), evidence is lacking regarding the associations between the three (i.e., resilience, stress management, and burnout).

*Spiritual growth* focuses on the development of internal resources through transcending (being in touch with the inner self), connecting (feeling a sense of harmony and wholeness with the world), and developing (searching for meaning and striving toward a goal; [39,40]. Earlier studies revealed that spiritual growth assists in coping with stressful events and increases the sense of well-being [39]. Spiritual growth and its connection to burnout has been studied as an overall concept and in relation to more specific examples, such as religion. In a study that examined an intervention designed to boost meaning, connecting, and spirituality in their influence on burnout, the intervention group reported less emotional exhaustion and a greater sense of personal accomplishment [40]. Other studies reported a negative correlation between spiritual growth and burnout [39,41]. In the literature search, we found no studies that addressed a direct association between resilience and spiritual growth. However, people

who reported higher resilience were found to perceive their moral beliefs as motives for their activities [42].

*Interpersonal relations* reflect the ability for intimate, meaningful communication with others while sharing thoughts and emotions through verbal and nonverbal channels. This concept has been widely researched in the literature in its broader context of social support, defined as others providing assistance and protection through informal or formal means [43]. A longitudinal study that examined medical students' satisfaction with their perceived social support found a decline in satisfaction during their first year of training [44]. In a different study, medical students in their clinical years reported feeling that they received better social support when discussing traumatic events with their professional peers than when talking to family members, friends, or intimate partners [45].

The interpersonal relations dimension has not been comprehensively studied in the context of burnout; nonetheless, a negative association was found between medical students' perceived social support and burnout [46]. Moreover, doctors who reported high emotional exhaustion expressed the wish to avoid being with people in general and not just in the workplace [1]. The prevailing assumption in the literature is that these two variables trigger each other: Difficulty with interpersonal relations leads to burnout, resulting in an additional deterioration in social functioning [18]. Finally, the concept of interpersonal relations in their limited context has not been extensively studied in the context of resilience.

*Health responsibility* relates to taking active measures to preserve one's physical and mental health: paying attention to one's health, health-related self-learning, and investing thought and consulting with experts in implementing health decisions [36]. Medical students are heavily occupied with health and comprehensive health-related self-learning. However, they were found to have a stronger tendency toward self-prescribing medication without consulting experts or registering in the medical system. This was possible through the receipt of prescriptions from trainees or specialist doctors with whom they worked [47]. One reason for this practice is that the medical system encourages medical staff to continue working, even when ill, rather than assuming the "patient role." Medical students reported that they avoid seeking professional treatment out of the fear that admitting to having a medical problem may jeopardize their academic status and compromise their medical confidentiality [48]. Hence the need for further study of the relationships between health responsibility, resilience, and burnout.

## The present study

Studies have addressed the resilience–burnout relationship (e.g., [13]) and the self-care–burnout relationship (e.g., [33]), but to the best of our knowledge, no studies to date have examined the three-way association of these variables and whether self-care mediates the associations between resilience and burnout among medical students. Therefore, this study examines these relationships.

## Hypotheses

H1: There will be linear associations between resilience and burnout: negative associations with exhaustion and depersonalization; positive associations with personal accomplishment.

H2: There will be linear associations between self-care and burnout: negative associations with exhaustion and depersonalization; positive associations with personal accomplishment.

H3: There will be positive linear associations between resilience and self-care.

H4: The associations between resilience and burnout will be mediated by self-care.

## Material and methods

### Sample and procedure

A quantitative cross-sectional study was conducted among a class of 95 fourth-year medical students from Tel Aviv University in Israel ($M_{age}$ = 27.26, $SD$ = 2.81), representing a response rate of 76% (95 of 125 students completed the questionnaires). Data from these students were collected in May 2016, immediately after a 14-week internal medicine clerkship. The clerkship is a critical and very intense stage in students' training, requiring them to adjust to the medical system. During this period, medical students may experience increased stress and distress [26]. The Tel Aviv University Ethics Committee approved the study (approval #6116), and all participants signed a written informed consent form before completing the self-reported questionnaires. (See Table 1 for participants' demographic characteristics and S1 File for the study fundamental data).

### Instruments

**Burnout.** This 9-item abbreviated Hebrew version [49] is based on the Maslach Burnout Inventory [50]. It measures burnout on three dimensions: emotional exhaustion (3 items, e.g., *"I feel used up at the end of the workday"*), depersonalization (3 items, e.g., *"I really don't care what happens to some patients"*), and personal accomplishment (3 items, *"I have accomplished many worthwhile things in this job"*). Items are rated on a 5-point Likert scale (1 = strongly disagree; 5 = strongly agree) and calculated by averaging the answers on each dimension, with higher scores on emotional exhaustion and depersonalization representing higher burnout, while higher scores on personal accomplishment represent lower burnout. Internal reliability in the present study was $\alpha$ = 0.827, $\alpha$ = 0.745, and $\alpha$ = 0.619, respectively.

**Resilience.** This 6-item Hebrew version is based on the Brief Resilience Scale (BRS) [51]. It measures individuals' ability to bounce back or recover from stress (e.g., *"I usually come through difficult times with little trouble"*). Items are rated on a 5-point Likert scale (1 = strongly disagree; 5 = strongly agree) and calculated by averaging the answers (after recoding three

**Table 1. Demographic characteristics of the study participants (N = 95).**

| | Category | Frequency | Valid Percent |
|---|---|---|---|
| **Gender** | Female | 60 | 63.2 |
| | Male | 35 | 36.8 |
| **Family status** | Single | 35 | 36.8 |
| | Relationship | 33 | 34.7 |
| | Married | 27 | 28.4 |
| **Religion**# | Jewish | 81 | 86.2 |
| | Muslim | 8 | 8.5 |
| | Christian | 3 | 3.2 |
| | Other | 2 | 2.1 |
| **Religiosity** | Secular | 74 | 77.9 |
| | Traditional | 11 | 11.6 |
| | Religious | 10 | 10.5 |

# In this variable there was one missing data. Hence, the total frequency sums in 94.

items), with higher scores representing higher resilience. Internal reliability in the present study was α = 0.879.

**Self-care.** This 27-item abbreviated version [52], translated into Hebrew for the present study, is based on the Health Promoting Lifestyle Profile II [40]. It measures self-care activities for the following dimensions: stress management (8 items; e.g., *"I balance time between work and play"*), spiritual growth (6 items; e.g., *"I feel I am growing and changing in positive ways"*), interpersonal relations (9 items; *"I maintain meaningful and fulfilling relationships with others"*), and health responsibility (4 items; *"I report any unusual signs or symptoms to a physician or other health professional"*). Items are rated on a 5-point Likert scale (1 = strongly disagree; 5 = strongly agree) and are calculated by averaging the answers on each dimension, with higher scores representing higher self-care. Internal reliability in the present study was α = 0.753, α = 0.795, α = 0.817, and α = 0.842, respectively.

## Statistical analysis

We used IBM-SPSS (version 25) to analyze the data. Pearson's correlations examined linear associations. PROCESS macro (version 3.3) examined mediation associations, using Conditional Process Analysis and calculating 5,000 bootstrapped samples, to estimate the 95% bias-corrected confidence intervals [53].

## Results

The present study examined the associations between resilience, self-care, and burnout among medical students (see Table 2 for the variables' psychometric characteristics and Pearson's correlations).

Table 2 demonstrates that Hypothesis 1, examining associations between resilience and burnout, was partially supported. We found a negative association regarding emotional exhaustion, i.e., students who reported high levels of resilience felt less exhausted. Additionally,

**Table 2. Psychometric characteristics and correlations of the study variables.**

|  | 1 | 2 | 3 | 4 | 5 | 6 | 7 | 8 |
|---|---|---|---|---|---|---|---|---|
| **1. Burnout: E** | --- | 0.24* | -0.27** | -0.29** | -0.35*** | -0.17 | -0.30** | 0.14 |
| **2. Burnout: D** |  | -- | -0.38*** | -0.03 | 0.04 | -0.14 | -0.15 | -0.08 |
| **3. Burnout: PA** |  |  | -- | 0.12 | -0.01 | 0.41*** | 0.29** | 0.00 |
| **4. Resilience** |  |  |  | -- | 0.23* | 0.41*** | 0.34*** | 0.04 |
| **5. Self-care: SM** |  |  |  |  | -- | 0.04 | 0.03 | 0.13 |
| **6. Self-care: SG** |  |  |  |  |  | -- | 0.47*** | 0.02 |
| **7. Self-care: IR** |  |  |  |  |  |  | -- | 0.20 |
| **8. Self-care: HR** |  |  |  |  |  |  |  | -- |
| *M* | 2.72 | 2.18 | 3.46 | 3.55 | 2.40 | 3.94 | 3.93 | 2.68 |
| *SD* | 0.88 | 0.78 | 0.68 | 0.80 | 0.55 | 0.54 | 0.48 | 0.92 |
| *Range* | 1–5 | 1–4.6 | 2–5 | 1–5 | 1.25–4.13 | 2–5 | 2.67–5 | 1–4.75 |
| *α* | 0.83 | 0.75 | 0.62 | 0.88 | 0.75 | 0.80 | 0.82 | 0.84 |

Burnout: E = exhaustion; D = depersonalization; PA = personal accomplishment.

Self-care: SM = stress management; SG = spiritual growth; IR = interpersonal relations; HR = health responsibility.

*$p<0.05$

**$p<0.01$

***$p<0.001$.

*M* = mean; *SD* = standard deviation; *Range* = minimum–maximum; *α* = Cronbach's alpha.

Hypothesis 2, examining associations between self-care and burnout, was partially supported. We found negative associations regarding emotional exhaustion, i.e., students who reported high stress management and interpersonal relations felt less exhausted. We also found a positive association regarding personal accomplishment, i.e., students who reported high spiritual growth and interpersonal relations had a greater sense of accomplishment. Finally, Hypothesis 3, examining associations between resilience and self-care, was partially supported. We found positive associations regarding three out of four activities, i.e., students who reported high resilience reported high stress management, spiritual growth, and interpersonal relations.

In addition to examining linear associations between the variables, we tested whether the associations between resilience and burnout are mediated by self-care (see below Fig 1 and Table 3). Fig 1 and Table 3 demonstrate that Hypothesis 4, examining the mediation model, was partially supported. In path a, examining the direct effects between resilience and self-care, we found positive associations regarding stress management, spiritual growth, and interpersonal relations. In path b, examining the direct effects between self-care and burnout, we found negative associations regarding emotional exhaustion (with the role of stress management and interpersonal relations) and a positive association regarding personal accomplishment (with the role of spiritual growth). Additionally, in path c, examining the total effects

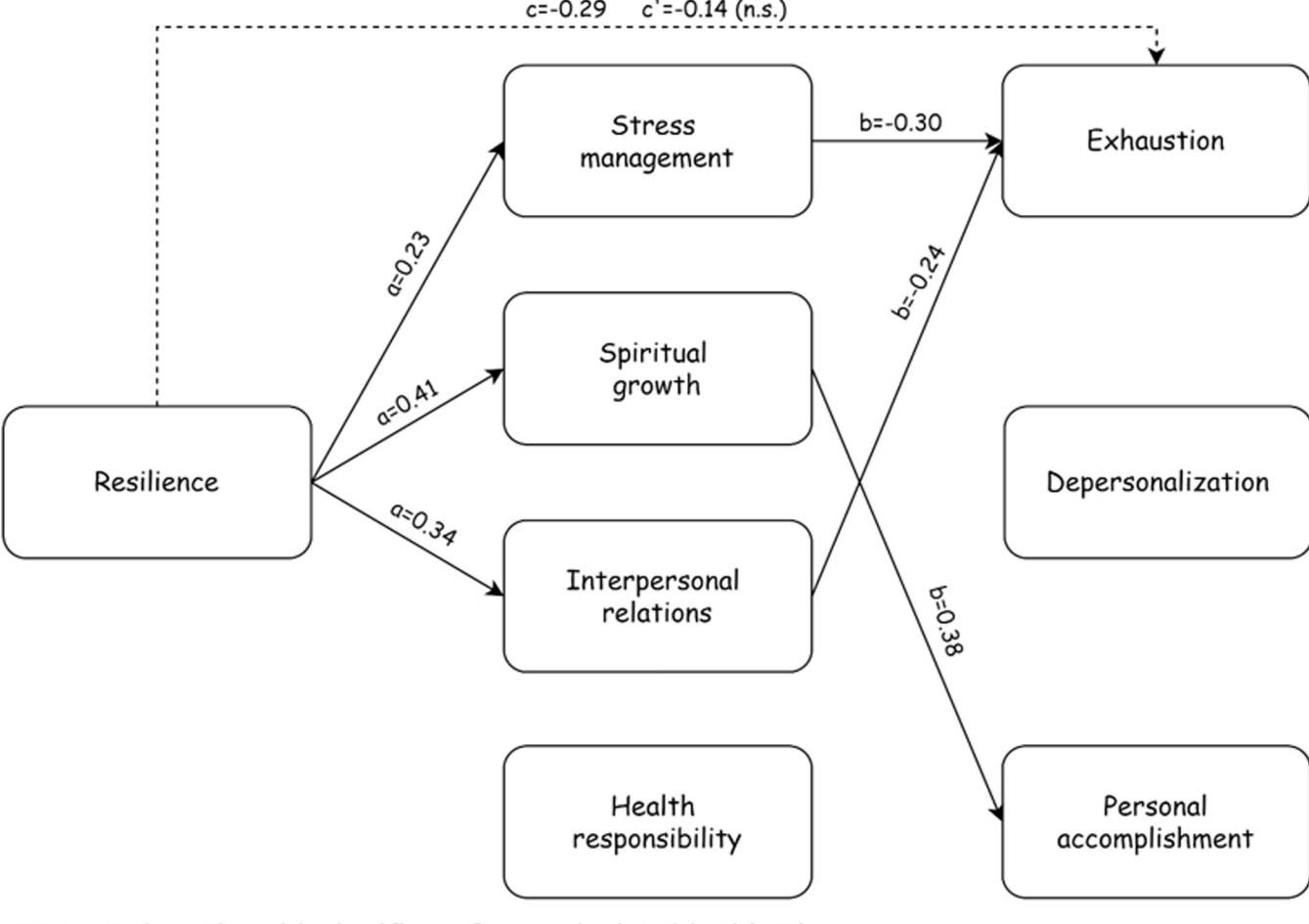

Note: Only paths with significant β were depicted in this Fig

**Fig 1. The study model: The association between resilience and burnout mediated by self-care.**

**Table 3. Regression analyses examining whether the association between resilience and burnout is mediated by self-care.**

| | Self-care: SM | | | | Self-care: SG | | | | Self-care: IR | | | | Self-care: HR | | | |
|---|---|---|---|---|---|---|---|---|---|---|---|---|---|---|---|---|
| | B | SE | β | LLCI ULCI | B | SE | β | LLCI ULCI | B | SE | β | LLCI ULCI | B | SE | β | LLCI ULCI |
| Resilience (path a) | 0.16 | 0.07 | 0.23 | 0.22 0.29 | 0.28 | 0.06 | 0.41 | 0.15 0.41 | 0.21 | 0.06 | 0.34 | 0.09 0.32 | 0.04 | 0.12 | 0.04 | -0.19 0.28 |

| | Burnout: E | | | | Burnout: D | | | | Burnout: PA | | | |
|---|---|---|---|---|---|---|---|---|---|---|---|---|
| | B | SE | β | LLCI ULCI | B | SE | β | LLCI ULCI | B | SE | β | LLCI ULCI |
| **Resilience (path c)** | -0.31 | 0.11 | -0.29 | -0.53 -0.10 | -0.03 | 0.10 | -0.03 | -0.23 0.17 | 0.10 | 0.09 | 0.12 | -0.08 0.28 |
| **Resilience (path c')** | -0.15 | 0.12 | -0.14 | -0.38 0.09 | 0.04 | 0.12 | 0.04 | -0.19 0.27 | -0.08 | 0.09 | -0.08 | -0.26 0.11 |
| **Self-care: SM (path b)** | -0.49 | 0.16 | -0.30 | -0.89 -0.17 | 0.06 | 0.15 | 0.05 | -0.24 0.37 | -0.00 | 0.13 | 0.00 | -0.25 0.25 |
| **Self-care: SG (path b)** | 0.01 | 0.18 | 0.01 | -0.35 0.37 | -0.15 | 0.18 | -0.11 | -0.51 0.20 | 0.47 | 0.14 | 0.38 | 0.19 0.76 |
| **Self-care: IR (path b)** | -0.44 | 0.20 | -0.24 | -0.84 -0.04 | -0.16 | 0.20 | -0.10 | -0.55 0.23 | 0.21 | 0.16 | 0.15 | -0.10 0.53 |
| **Self-care: HR (path b)** | -0.04 | 0.09 | -0.04 | -0.22 0.14 | -0.05 | 0.09 | -0.06 | -0.23 0.13 | -0.02 | 0.07 | -0.03 | -0.17 0.12 |
| **Resilience> Self-care: SM (path ab)** | -0.08 | 0.04 | -0.07 | -0.18 -0.12 | 0.01 | 0.03 | 0.01 | -0.04 0.07 | 0.00 | 0.02 | 0.00 | -0.04 0.53 |
| **Resilience> Self-care: SG (path ab)** | 0.00 | 0.05 | 0.00 | -0.09 0.11 | -0.04 | 0.05 | -0.04 | -0.16 0.06 | 0.13 | 0.06 | 0.15 | 0.04 0.28 |
| **Resilience> Self-care: IR (path ab)** | -0.09 | 0.07 | -0.08 | -0.27 -0.01 | -0.03 | 0.06 | -0.04 | -0.17 0.06 | 0.04 | 0.04 | 0.05 | -0.02 0.14 |
| **Resilience> Self-care: HR (path ab)** | -0.00 | 0.01 | -0.00 | -0.03 0.03 | -0.00 | 0.01 | -0.00 | -0.04 0.02 | -0.00 | 0.09 | -0.00 | -0.02 0.02 |

Self-care: SM = stress management; SP = spiritual growth; IR = interpersonal relations HR = health responsibility.

Burnout: E = exhaustion; D = depersonalization; PA = personal accomplishment.

B = unstandardized beta; *SE* = standard error for the unstandardized beta; β = standardized beta; *T* = *t*-test statistic.

*LLCI–ULCI* = lower limit of the confidence interval–upper limit of the confidence interval.

between resilience and burnout, we found a positive association regarding emotional exhaustion. As expected, it becomes insignificant while controlling this total effect with the other variables (path c'). Finally, in paths ab, examining indirect effects between resilience and burnout, we found mediations of stress management and interpersonal relations between resilience and emotional exhaustion, and mediation of spiritual growth between resilience and personal accomplishment.

## Discussion

The present study examined associations between resilience, self-care, and burnout among medical students at the beginning of their clinical training, a period that evokes acute stress and burnout [26]. The study hypotheses were partially supported since associations between resilience, self-care, and burnout were found, but not in all self-care and burnout dimensions. The study findings demonstrated associations between the variables insofar as the self-care variable mediated the association between resilience and burnout for some of the self-care and burnout dimensions.

Like earlier studies, the present study showed that resilience has a direct negative association with one burnout dimension—emotional exhaustion [54–57]. This finding is consistent with the theory of positive psychology [58], which asserts that when individuals succeed in overcoming stressful situations, they undergo an internal change and a process of growth that allows them to moderate the emotional exhaustion and to avoid the sense of emptiness, fatigue, and lack of motivation. Furthermore, in a qualitative study among resilient teachers, most participants reported that their ability to grow and learn from previous experiences with difficult situations helped them avoid a sense of emptiness, fatigue, and stress. This ability also led to a sense of control and responsibility for their actions as significant components of their

resilience [59]. Individuals who have this sense of control will often perform well-being-enhancing actions, such as engaging in self-care practices.

Indeed, the present study model broadened this understanding, showing that the association between resilience and emotional exhaustion is mediated by two self-care practices: stress management and interpersonal relations. These findings are in accordance with Strümpfer's [21] theory that claims that proactive coping, which relates to people's ability to identify stress-inducing situations and to behave in a way that will prevent their aggravation and will evoke a sense of control and self-efficacy, is one of the resilience components. The findings of the present study suggest that effective stress management is a behavior that may prevent the worsening of stressful situations. In that way, students with high resilience can predict situations that may cause stress (e.g., academic overload before the internal medicine clerkship exam) and therefore implement a suitable coping strategy; namely stress management (e.g., practicing calming techniques at different hours of the day) or seeking social support and interpersonal relations as a buffer and as protection from the negative influences of the cause of stress [60].

Regarding depersonalization, the interpersonal relations dimension that emphasizes personal connection might have been expected to help reduce burnout, since meaningful relationships with others would decrease negative perceptions of patients or the act of relating to people as objects [1]. However, this association was not found to be significant in the present study. One reason for this is that we had no information about whom the study participants had interpersonal relationships with. Their interpersonal relationships may have been primarily with student peers—a population on whom medical students tend to rely for sharing their experiences of traumatic events [45]. If that was the case, these relationships might be part of the professional socialization process, in which the hospital's organizational culture and norms encourage emotional detachment [61]. Therefore, these relationships, despite their advantage in providing social support and protection from emotional exhaustion, do not contribute to the prevention of depersonalization and cynicism.

The hospital norms can also explain why other study variables were not found to be directly or indirectly associated with depersonalization as a dimension of burnout. Depersonalization variables exist in the organizational culture, and their origins are deeper and more socially based [61]. One example is the norm that doctors do not cry. Therefore, they do not encourage the spontaneous expression of emotions [61] and often, do not behave empathetically [62]. Thus, this aspect is less dependent on the individual's development (e.g., increased resilience or self-care), and other buffering/protecting factors require enhancement, such as encouragement and teaching of empathy [63].

Regarding personal accomplishment, no direct association was found between resilience and this burnout dimension, but an indirect association was found between them, mediated by spiritual growth. According to existentialist theory, people need a sense of meaning in life and the belief that the actions they perform have a generally positive and even heroic effect [21]. In the absence of a sense of a mission and influence in the workplace, people experience burnout due to the lack of personal accomplishment. According to positive psychology, resilient individuals will find meaning and a sense of a mission in stressful situations and in their ability to overcome them [58]. Finding meaning and a sense of mission contribute to spiritual growth: development of internal resources by transcending (being tuned to the inner self), connecting (a sense of harmony and wholeness), and developing (including a search for meaning and striving toward a goal; [39–40]. Transcending, connecting, and developing help build a positive self-perception and thus increase self-accomplishment [41].

Health responsibility, relating to the active preservation of the individual's physical and mental health [36], was examined in the present study since it is documented in the literature as problematic among health professionals [64]. As far as we know, the present study is the

first to examine the associations between resilience, health responsibility, and burnout. Our findings, which indicate an absence of association between the variables, are interesting and can be explained by the fact that health responsibility, as expressed in the questionnaire, relates to physical rather than mental health and, therefore, might not be associated with burnout dimensions. Another explanation may relate to the fact that physical health responsibility was less relevant to our students because of their relatively young age. Most of the students (93%) were below the age of 30. Even students who neglect their health and do not consult specialist physicians are not yet suffering from the harmful impact of that neglect on their daily functioning.

Alongside the specific findings explained above, self-care was a mediator between resilience and burnout. That is to say that resilience leads to a coping strategy that creates protection from burnout. The explanation for this process may be grounded in seeing resilience as a developmental issue: an acquired capability influenced by individual or group traits and experiences that affect different outcomes in people's behavior, feelings, and well-being [65]. This raises the question of which experiences are required to develop resilience. According to the behavioral immunization theory [58], to become immune, the body must encounter pathogens (e.g., bacteria or a virus) and create antibodies to destroy them. After the initial illness, the immune system's memory retains the antibodies and prevents reinfection. Similarly, to develop resilience, the person needs to "encounter" a stressful situation (e.g., the death of a patient) and to recognize ways to cope with it (e.g., debriefing this experience with a mentor or close friend). Effective coping will be retained in the individual's "memory" and will be a platform for increasing the sense of control in similar stress situations in the future.

In this manner, the findings of the present study suggest that self-care is, in fact, a coping strategy. In stressful situations, people learn to identify activities that reduce stress. Individuals with high resilience have coping strategies, such as stress management, spiritual growth, and interpersonal relations. When they experience a stressful situation, they identify the need to use the strategy and are, therefore, less prone to burnout. However, individuals with low resilience may have difficulty identifying helpful coping strategies and have a lower tendency toward self-care activities. Therefore, they are more at risk of burnout and its negative implications.

Nonetheless, self-care behaviors that are disconnected from the stress and coping context will not necessarily lead to resilience. This was tested in an alternative model of the present study, in which resilience did not mediate the association between self-care and burnout. This finding can be explained in the context of the organizational culture of medicine. The norm in hospitals is a dichotomic separation between caregivers and patients: a clear distinction between providers and receivers of help. Therefore, when entering clinical rounds, many students feel that requesting help or choosing to perform self-care activities goes against the "tough" norm [18,48]. In this way, and in accordance with the behavioral immunization theory [58], students with low resilience will either have difficulty developing resilience under these norms or will develop coping strategies that are appropriate to the existing norms, including neglect of personal life [18], cynicism toward the environment, and other methods that lead to burnout. In contrast, students with high resilience will react to stressful situations according to their "immune memory" and, during stressful situations, will adopt familiar, helpful self-care behaviors, based on the understanding that they will enable them to function better in their work.

## Implications

The findings of the present study have significant implications for planning medical students' curricula. Today, medical schools invest time and resources in developing various burnout

prevention interventions to promote students' well-being [66]. Most of these interventions emphasize encouraging self-care behaviors, such as healthy nutrition, leisure activities, stress-reducing workshops, and socializing with others [18]. The findings of the present study support the behavioral immunization theory [58], according to which, first, the development of resilience must be encouraged while experiencing mild to moderate stress situations; and then, while in these situations, learning effective coping strategies, such as self-care techniques. This awareness leads students to rely on these behaviors and help prevent burnout when encountering challenging situations in the future. Increasing resilience includes focusing on self-efficacy, self-regulation, positive thinking, flexible thinking, emotional awareness, and problem-solving ability—and interpersonal components, such as developing empathy toward others and strengthening relationships [67]. These programs focus on a cognitive change in perception of the stressful situation. They help to develop strategies for coping with the stress by offering examples of possible coping methods and relevant daily-life situations, inter alia, through simulations and role-plays [68]. Based on the results of the present study, applying such programs in medical schools can help develop resilience and thereby reduce burnout. However, additional research is needed to establish causal associations.

## Limitations

Alongside the important implications of the study, various limitations must be considered. The sample chosen for this study does not necessarily represent all medical students. All these students completed the questionnaire in the same week, at an identical time point, after completing 14 weeks of their first clerkship—in internal medicine. This limitation threatens the external consistency of the study due to experiences unique to this university: all participants did their clinical clerkships in large hospitals in the center of the country, took part in interventions within the framework of communication and professionalism courses [69,70], and were exposed to the same norms and experiences as each other [62]. Another limitation is the research design: We conducted a cross-sectional study using statistical techniques to examine a mediating model. No additional intervention or measurement was performed afterward; therefore, we cannot deduce causal associations from its results.

## Conclusions

Burnout among medical students is a global problem with far-reaching implications for students' physical and mental health and the health of their future patients. The present study suggests that medical students' high levels of resilience are associated with taking more self-care actions (stress management, spiritual growth, and interpersonal relations). Additionally, taking more self-care actions is associated with lower levels of burnout (emotional exhaustion and personal accomplishment). The findings of the present study recommend transferring the emphasis in medical schools to developing resilience resources among medical students while exposing them to self-care activities in this context. Furthermore, we recommend exploring other interventions in the attempt to prevent the critical burnout dimension of depersonalization that goes beyond resilience and self-care.

## Supporting information

**S1 File. The study fundamental data.**
(PDF)

## Author Contributions

**Conceptualization:** Keren Michael, Orit Karnieli-Miller.

**Formal analysis:** Keren Michael, Dana Schujovitzky.

**Investigation:** Keren Michael, Orit Karnieli-Miller.

**Supervision:** Keren Michael, Orit Karnieli-Miller.

**Writing – original draft:** Keren Michael, Dana Schujovitzky, Orit Karnieli-Miller.

**Writing – review & editing:** Keren Michael, Dana Schujovitzky, Orit Karnieli-Miller.

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
