## [Decision Letter · Decision Letter 0]

17 Jul 2024

PONE-D-24-16703The Associations Between Resilience, Self-Care, and Burnout among Medical StudentsPLOS ONE

Dear Dr. Michael,

Thank you for submitting your manuscript to PLOS ONE. After careful consideration, we feel that it has merit but does not fully meet PLOS ONE’s publication criteria as it currently stands. Therefore, we invite you to submit a revised version of the manuscript that addresses the points raised during the review process.

We look forward to receiving your revised manuscript.

Kind regards,

Douglas Aninng Opoku, MPH

Academic Editor

PLOS ONE

3. hank you for stating the following financial disclosure: 

 [THE ISRAEL NATIONAL INSTITUTE FOR HEALTH POLICY RESEARCH partially supported the study.].  

5. Please include your tables as part of your main manuscript and remove the individual files. Please note that supplementary tables (should remain/ be uploaded) as separate ""supporting information"" files.

6. We are unable to open your Supporting Information file [self care-PO.sav]. Please kindly revise as necessary and re-upload.

Comments from PLOS Editorial Office: We note that one or more reviewers has recommended that you cite specific previously published works. As always, we recommend that you please review and evaluate the requested works to determine whether they are relevant and should be cited. It is not a requirement to cite these works. We appreciate your attention to this request.

Reviewers' comments:

Reviewer's Responses to Questions

**Comments to the Author**

1. Is the manuscript technically sound, and do the data support the conclusions?

Reviewer #1: Yes

Reviewer #2: Partly

2. Has the statistical analysis been performed appropriately and rigorously? 

Reviewer #1: Yes

Reviewer #2: I Don't Know

3. Have the authors made all data underlying the findings in their manuscript fully available?

Reviewer #1: Yes

Reviewer #2: Yes

4. Is the manuscript presented in an intelligible fashion and written in standard English?

Reviewer #1: Yes

Reviewer #2: Yes

5. Review Comments to the Author

Reviewer #1: Dear Author(s),

I read your work with great interest, and I am pleased to congratulate you on your contribution to scientific research.

I believe that the article is novel and interesting, that it has a sufficient impact, and that it adds to the knowledge base. Plagiarism was not detected. The study appears to follow relevant guidelines and provides an original contribution to the existing scientific literature. There are no flaws in the data presented, and there are no misleading or false conclusions.

The current study is scientifically valid. The reasons for performing the study are clear. However, before proceeding with its publication, it is necessary to make some minor revisions. I strongly suggest that the authors highlight that resilience has been proven to be key to the mental health and well-being of healthcare workers (please cite: Safiye, T., Vukčević, B., Čabarkapa, M. (2021). Resilience as a moderator in the relationship between burnout and subjective well-being among medical workers in Serbia during the COVID-19 pandemic. Vojnosanitetski Pregled, 78 (11), 1207–1213.).

I recommend the article for publication in PLOS ONE after minor revisions.

Sincerely,

Reviewer

Reviewer #2: Summary: A well written manuscript impressive flow of thought and written in intelligible fashion

Results: Table 1: the columns do not correspond correctly with the other categories in the rows. Please see highlighted portions of the table

Introduction and methods: some highlighted portions require rephrasing

Data which was used for the regression analysis was not accessed by me

Conclusion: 'The present study reveals that strengthening medical students' resilience levels is related to encouraging self-care actions (stress management, spiritual growth, and interpersonal relations) and is, in turn, related to a lower level of burnout (emotional exhaustion and personal accomplishment).'

comment: Given the study design it is difficult to agree with this comment

References 1-4 should be written in full in consistency with all other references

6. PLOS authors have the option to publish the peer review history of their article (what does this mean?). If published, this will include your full peer review and any attached files.

Reviewer #1: No

Reviewer #2: No

---

## [Author Response · Author response to Decision Letter 0]

23 Jul 2024

Thank you very much for the supportive review and guidance

---

## [Decision Letter · Decision Letter 1]

20 Aug 2024

PONE-D-24-16703R1The Associations Between Resilience, Self-Care, and Burnout among Medical StudentsPLOS ONE

Dear Dr. Michael,

Thank you for submitting your manuscript to PLOS ONE. After careful consideration, we feel that it has merit but does not fully meet PLOS ONE’s publication criteria as it currently stands. Therefore, we invite you to submit a revised version of the manuscript that addresses the points raised during the review process.

Both reviewers were very positive about your work particularly on how you addressed their comments that were raised in their last review. However, there are a few grammatical issues raised by one of the reviewers before your work can be ready for publication. 

We look forward to receiving your revised manuscript.

Kind regards,

Douglas Aninng Opoku, MPH

Academic Editor

PLOS ONE

Journal Requirements:

Reviewers' comments:

Reviewer's Responses to Questions

**Comments to the Author**

1. If the authors have adequately addressed your comments raised in a previous round of review and you feel that this manuscript is now acceptable for publication, you may indicate that here to bypass the “Comments to the Author” section, enter your conflict of interest statement in the “Confidential to Editor” section, and submit your "Accept" recommendation.

Reviewer #1: All comments have been addressed

Reviewer #2: All comments have been addressed

2. Is the manuscript technically sound, and do the data support the conclusions?

Reviewer #1: Yes

Reviewer #2: Yes

3. Has the statistical analysis been performed appropriately and rigorously? 

Reviewer #1: Yes

Reviewer #2: I Don't Know

4. Have the authors made all data underlying the findings in their manuscript fully available?

Reviewer #1: Yes

Reviewer #2: Yes

5. Is the manuscript presented in an intelligible fashion and written in standard English?

Reviewer #1: Yes

Reviewer #2: Yes

6. Review Comments to the Author

Reviewer #1: Dear Authors,

The manuscript has been improved, and I recommend that the paper be accepted without any changes and published in the journal PLOS ONE.

Best regards

Reviewer #2: The statistical analysis seems to have been performed appropriately however I did not replicate with the data to determine the accuracy

Suggested Corrections

Introduction: Page 2, line 8 – “comprising their own well-being” should read “compromising their own well- being”

Page 3 line 11 - “underlying” can be better put: suggestion: “underpinning”

Discussion: Page 15, lines 12 and 13: “with whom the present study participants had interpersonal relationships” should be rephrased: suggestion: “whom the study participants had interpersonal relationships with”

Page 18, line 10 “in their stressful lives, will include”, suggestion: “during stressful situations will adopt”

7. PLOS authors have the option to publish the peer review history of their article (what does this mean?). If published, this will include your full peer review and any attached files.

Reviewer #1: No

Reviewer #2: No

---

## [Author Response · Author response to Decision Letter 1]

21 Aug 2024

Dear Editor,

Thank you for the opportunity to revise and resubmit our manuscript. We greatly appreciate the time and effort you and the reviewers invested in evaluating our manuscript and providing constructive feedback.

We are pleased that the reviewers responded positively to our revisions in response to their previous comments. We followed Reviewer 2 grammatical concerns and corrected our manuscript accordingly.

---

## [Editor Report · Decision Letter 2]

23 Aug 2024

The associations between resilience, self-care, and burnout among medical students

PONE-D-24-16703R2

Dear Dr. Michael,

We’re pleased to inform you that your manuscript has been judged scientifically suitable for publication and will be formally accepted for publication once it meets all outstanding technical requirements.

Kind regards,

Douglas Aninng Opoku, MPH

Academic Editor

PLOS ONE
---

## [Editor Report · Acceptance letter]

10 Sep 2024

PONE-D-24-16703R2 

PLOS ONE

Dear Dr. Michael, 

I'm pleased to inform you that your manuscript has been deemed suitable for publication in PLOS ONE. Congratulations! Your manuscript is now being handed over to our production team.

Kind regards, 

on behalf of

Dr. Douglas Aninng Opoku 

Academic Editor

PLOS ONE